# *Doratoxylon apetalum,* an Indigenous Medicinal Plant from Mascarene Islands, Is a Potent Inhibitor of Zika and Dengue Virus Infection in Human Cells

**DOI:** 10.3390/ijms20102382

**Published:** 2019-05-14

**Authors:** Juliano G. Haddad, Andrea Cristine Koishi, Arnaud Gaudry, Claudia Nunes Duarte dos Santos, Wildriss Viranaicken, Philippe Desprès, Chaker El Kalamouni

**Affiliations:** 1Université de la Réunion, INSERM U1187, CNRS UMR 9192, IRD UMR 249, Unité Mixte Processus Infectieux en Milieu Insulaire Tropical, Plateforme Technologique CYROI, 94791 Sainte Clotilde, La Réunion, France; juliano.haddad@univ-reunion.fr (J.G.H.); wildriss.viranaicken@univ-reunion.fr (W.V.); philippe.despres@univ-reunion.fr (P.D.); 2Laboratorio de Virologia Molecular, Instituto Carlos Chagas, ICC/FIOCRUZ/PR, Curitiba, Parana 81350-010, Brazil; ackoishi@gmail.com (A.C.K.); clsantos@fiocruz.br (C.N.D.d.S.); 3Bioactive natural products unit, School of Pharmaceutical Sciences, EPGL, University of Geneva, Rue Michel-Servet 1, CH-1211 Geneva, Switzerland; Arnaud.gaudry@unige.ch

**Keywords:** Zika virus, dengue virus, antiviral activity, medicinal plant, nutraceuticals, polyphenol, *Doratoxylon apetalum.*

## Abstract

Zika virus (ZIKV) and Dengue virus (DENV) are mosquito-borne viruses of the *Flavivirus* genus that could cause congenital microcephaly and hemorrhage, respectively, in humans, and thus present a risk to global public health. A preventive vaccine against ZIKV remains unavailable, and no specific antiviral drugs against ZIKV and DENV are licensed. Medicinal plants may be a source of natural antiviral drugs which mostly target viral entry. In this study, we evaluate the antiviral activity of *Doratoxylum apetalum*, an indigenous medicinal plant from the Mascarene Islands, against ZIKV and DENV infection. Our data indicated that *D. apetalum* exhibited potent antiviral activity against a contemporary epidemic strain of ZIKV and clinical isolates of four DENV serotypes at non-cytotoxic concentrations in human cells. Time-of-drug-addition assays revealed that *D. apetalum* extract acts on ZIKV entry by preventing the internalisation of virus particles into the host cells. Our data suggest that *D. apetalum*-mediated ZIKV inhibition relates to virus particle inactivation. We suggest that *D. apetalum* could be a promising natural source for the development of potential antivirals against medically important flaviviruses.

## 1. Introduction

Zika virus (ZIKV) and the closely related dengue virus (DENV) are mosquito-borne RNA viruses of the genus *Flavivirus* in the *Flaviviridae* family [1]. ZIKV has recently emerged as a public health threat because of its rapid spread in the world, sexual transmission and vertical human-to-human transmission, and its association with congenital malformations and neurological disorders [2,3,4]. The disease burden due to four dengue serotypes (DENV 1–4) has recently been revised, and accordingly, about 390 million infections occur annually in over 100 countries in tropical and sub-tropical regions of the world [5].

Flavivirus genomic RNA consists of a single ORF, translated into a polyprotein, and processed to yield 3 structural (capsid [C], premembrane [prM] and envelope [E]) and 7 nonstructural (NS1 to NS5) proteins which play an important role in the replication of the virus [6]. After the binding of virus particles to the cell surface receptors, their internalisation can occur through a clathrin-dependent pathway. The structural rearrangement of the E protein mediates virus-to-cell membrane fusion, releasing viral nucleocapsid into the cytosol. The fusion of the viral envelope with the endosomal membrane requires a low-pH environment [7,8].

At present, there are no therapeutics licensed against ZIKV and DENV infections. Given that many countries worldwide remain at risk of ZIKV and DENV outbreaks due to the prevalence of *Aedes* spp. vectors [5,9,10], it is of utmost urgency to develop safe and effective antivirals by exploring the potential of medicinal plants as natural sources of nutraceuticals that could be used to prevent virus infection [11,12,13,14,15,16,17]. The use of polyphenol-rich medicinal plants and their purified compounds as potential antiviral therapies has been recently explored [11,12,15,18,19,20,21,22,23,24]. Several phytochemical families including polyphenols, flavonoids, alkaloids and curcuminoids have been reported to inhibit flavivirus infection [18,21,22,24,25,26,27]. It has been demonstrated that epigallocatechin gallate (EGCG) from green tea, isoquercitrin (Q3G) and curcumin impair ZIKV and DENV infection [17,18,21,28,29,30]. We described that EGCG acts on the early stage of ZIKV and DENV infection by inhibiting the binding of the virus to the cell surface [12,24,29], whereas Q3G inhibits the internalisation process of ZIKV infection in human cells [22]. Recent studies have demonstrated that a polyphenol-rich extract from *Doratoxylon apetalum*, an indigenous medicinal plant from the Mascarene Islands, was able to protect cells from oxidative stress due to its high polyphenolic content [31,32]. In the present study, we showed that *D. apetalum* extract exerts an antiviral effect against ZIKV and DENV in human cells.

## 2. Results and Discussion

### 2.1. D. apetalum is an Effective Suppresor of ZIKV Infection at Non-Cytotoxic Concentration

*Doratoxylon apetalum* extract was first tested on Vero, A549, and Huh7.5 cells for cytoxicity using a 3-[4,5-dimethylthiazol-2-yl]-2,5- diphenyltetrazolium bromide (MTT) assay. D. apetlum extract showed little or no cytotoxicity at concentrations < 200 µg·mL^−1^ regardless of the cell lines tested (Figure 1). Huh7.5 cells were the most sensitive for *D. apetalum* extract. The 50% cytotoxic concentration (CC_50_) were 1250, 560 and 350 µg·mL^−1^ for Vero, A549 and Huh7.5 cell lines, respectively (Figure 1). Therefore, various concentrations of plant extract up to 200 µg·mL^−1^ were used in the futher experiments.

Given that human lung epithelial A549 cells support infection by the epidemic strain PF-25013-18 (PF13) of ZIKV, PF13-infected A549 cells were used to evaluate the anti–ZIKV activity of *D. apetalum* extract. By immunofluorescence analysis using anti-flavivirus E mAb 4G2, we showed that *D. apetalum* extract severely restricted ZIKV infection in A549 cells yielding a 75% inhibition of ZIKV infection at 50 µg·mL^−1^ (Figure 2A). Likewise, viral protein production was severely inhibited in a concentration-dependent manner (Figure 2B). At the higher non-cytotoxic concentration of 100 µg·mL^−1^, *D. apetalum* extract reduced the viral progeny production by at least 5 log_10_ (Figure 2C). No viral growth was detected at 200 µg·mL^−1^ of *D. apetalum* extract, indicating that such a concentration provided near complete protection against PF13 infection. Altogether, these data demonstrate that *D. apetalum* extract can efficiently inhibit ZIKV infection in A549 cells in a dose-dependent manner and reflect its potential as a source of natural antiviral phytochemicals.

### 2.2. D. apetalum Extract Inhibits Infection by Four DENV Serotypes

We next wondered if *D. apetalum* extract has a potential antiviral activity against DENV, another medically interesting flavivirus. Thus, the antiviral activity of *D. apetalum* extract was evaluated using four different clinical isolates representing four DENV serotypes. IFN-α 2A, which is known to block DENV replication [12], was used as positive control. Huh7.5 cells were inoculated with four DENV serotypes in the presence or absence of different non-cytotoxic concentrations of *D. apetalum* extract. The number of infected cells were quantified at 48 h post-infection (h.pi) by immunofluorescence assay using the Operetta high-content imaging system [12]. Our data demonstrate that *D. apetalum* extract exerts an antiviral effect against the four DENV serotypes and epidemic Brazilian strain ZIKV-BR of ZIKV in a dose-dependent manner (Figure 3). The 50% inhibitory concentration (IC_50_) values were 96.35, 16.75, 25.90, 23.30, and 17.50 µg·mL^−1^ for DENV1-4 and ZIKV-BR, respectively. DENV-2 appears to be the most sensitive virus. The selectivity index (SI), which measures the preferential antiviral activity of a drug in a relation to its cytotoxicity [22], was calculated according to their CC_50_ and IC_50_ established using non-linear regression curve (Appendix A). The SIs were 2.7, 17.8, 11.3, 13.0 and 16.8 for DENV1-4 and ZIKV-BR, respectively (Table 1). DENV-1 was the least sensitive virus against *D. apetalum* extract.

### 2.3. D. apetalum Extract-Mediated Inhibition of ZIKV Occurs at the Early Stage of the Viral Infetious Cycle

To further characterize the antiviral mechanism and the stage of ZIKV infection affected by *D. apetalum* extract, we examined the impact of the plant extract on various stages of the dissected virus replication cycle (pre-entry, entry and post-entry). In addition, wash steps were included to ensure the specificity of the treatment on the distinct stage examined [22,33]. In order to study pre-entry—more specifically, the ability of *D. apetalum* to induce host cell innate immunity prior virus addition [34]—A549 cells were pretreated with *D. apetalum* (200 µg·mL^−1^) for 6h (Figure 4A: pre-treatment). For effects on the viral entry stage, ZIKV and *D. apetalum* extract were simultaneously co-added to the cells during 2h (Figure 4A: coaddition). To investigate events after virus entry, A549 cells were first infected with ZIKV for 2 h and then treated with *D. apetalum* (Figure 4A: post-infection). For comparison, *D. apetalum* extract was also maintained throughout the experimental period. Pretreating A549 cells with *D. apetalum* extract slightly protects against ZIKV infection (*p* < 0.05). Compared to the pretreatment of the cells with *D. apetalum* extract or to adding it in the entry stage or post-infection phase, the anti-ZIKV effect was most evident during the coaddition treatment, yielding a significant reduction in GFP-positive cells (90% of inhibition, *p* < 0.001) (Figure 4B). These results therefore suggest that the antiviral activity of *D. apetalum* extract are unlikely to be mediated by effects through triggering antiviral innate immunity or viral replication. All together, these data suggest that *D. apetalum* targets the initial stages of ZIKV infection, whereby both virus and plant extract are present on the cell surface.

### 2.4. D. apetalum Extract Prevents ZIKV Entry by Inactivating Virus Particles

To further elucidate the underlying mechanism, we subsequently tested the influence of D. apetlaum on specific ZIKV entry steps. We first investigated whether *D. apetalum* directly affects the cell-free virions to abolish subsequent infection (Figure 5A) [12,34]. For this, ZIKV particles were pre-incubated with *D. apetalum* extract for 2 h and then diluted 50-fold prior to infection. This dilution titrates the plant extract below its therapeutic concentration and prevents potential interactions with the host cell surface (Figure 5A). Flow cytometry and viral titration assays showed that ZIKV infectivity was severely affected by *D. apetalum* extract yielding a 90% inhibition of infection as well as 3 log_10_ of viral progeny reduction (Figure 5A). Thus, *D. apetalum* extract could irreversibly interact with virus particules to prevent infection. Altogether, these results suggest that phytochemicals present in *D. apetalum* extract could bind to virus particles and neutralize virus infectivity.

We next assessed the ability of *D. apetalum* extract to affect viral attachment. The attachment assays were performed at 4 °C, which allows for virus binding but prevents viral entry (Figure 5B) [35]. The polyphenol EGCG, which is known to inhibit ZIKV attachment [12,19,24], was used as a positive control. After 1 h of ZIKV attachment with or without *D. apetalum*, the number of attached ZIKV particles was evaluated using RT-qPCR (Figure 5B). Our data show that the number of viral particles attached to the cell surface membrane was similar to untreated cells. As expected, EGCG significantly inhibits ZIKV attachment to the cell membrane (Figure 5B). The absence of difference in the amount of viral particles attached to the cell membrane compared to untreated cells suggests that the incapacity of ZIKV to initiate productive infection in the presence of *D. apetalum* extract was not related to a defect in cell-attachment (Figure 5B). Altogether, the results showed that *D. apetalum* extract prevents ZIKV entry in the host cell without affecting the attachment step.

Our previous antiviral assays showed that the *D. apetalum* extract-mediated inhibition of ZIKV is related to a virus inactivation affecting post-attachment step(s) in the infectious virus cycle. To further assess the effect of *D. apetalum* extract on the virus internalisation step, ZIKV particles were allowed to bind to A549 cells at 4 °C, without *D. apetalum* treatment, and were subsequently internalized into the host cell by shifting the temperature to 37 °C in the presence of *D. apetalum* extract (Figure 5C). Isoquercitrin (Q3G), which is known to inhibit the ZIKV internalisation process in A549 cells, was used as a positive control [22]. Both *D. apetalum* and Q3G showed a similar time-course of ZIKV entry inhibition with a maximum of effect during the first 30 min post-binding (Figure 5C). This result suggests that *D. apetalum* has the ability to inhibit the early steps of virus internalisation into the host cell.

Altogether, these data provide evidence that *D. apetalum* extract-mediated inhibition of ZIKV infection occurs early after virus binding to the plasma membrane and could be explained by the inability of the plasma membrane-associated virus to be internalized into the host cell in presence of plant extract.

### 2.5. Concluding Remarks

Currently, there is no specific treatment available against ZIKV and DENV infections [5,9]. Therefore, the development of natural substances as nutraceuticals able to inhibit virus infection represents an attractive preventive strategy [11,36]. In this field, medicinal plants may be a source of promising natural antiviral compounds against emerging arboviruses [11,12,17,18,21,22,27,29,30,37]. In this study, we report that an extract from *Dorotoxylon apetalum*, an indigenous medicinal plant from the Mascarene Islands, inhibits infection of A549 and Huh7.5 cells by epidemic ZIKV and DENV strains without reducing cell viability. In addition, our results suggest that *D. apetalum* extract could bind to ZIKV particles, rendering the virus incapable of initiating an infectious viral cycle. Pretreatment of host cells with *D. apetalum*, followed by washes to remove unadsorbed phytochemicals, had a slight effect upon ZIKV infection. That indicated that masking cell surface receptors or entry factors for ZIKV is likely [34]. Viral binding assays using RT-qPCR reveals that *D. apetalum* extract does not interfere with viral attachment to the host cell membrane. While *D. apetalum* extract could inactivate the ZIKV particles, we do not believe that a direct lysis effect of the viral membrane is responsible for their effects, since ZIKV attachment to the cell membrane can occur with *D. apetalum*. Future experiments using confocal microscopy will be undertaken to validate our assumption that *D. apetalum*-mediated ZIKV inhibition essentially relates to a lack of virus internalisation into the host cell [38].

Our data demonstrate that *D. apetalum*-mediated flavivirus inhibition is linked to a severe loss of viral infectivity and a lack of virus internalisation into the host cells. Whether a single or several phytochemical(s) can contribute to the antiviral action of *D. apetalum* is a critical issue that remains to be investigated. It is also a high priority to determine which viral factors are targeted by the phytochemical(s) present in *D. apetalum*. The flavivirus E protein plays a key role in the recognition of the flavivirus receptors at the cell surface and the subsequent internalisation of virus particles into the endosomal compartment where viral membrane–cell membrane fusion can take place, releasing the viral genome into the cytosol [1,6,7,8,39,40]. It would be of interest to evaluate whether E protein is the main target of *D. apetalum*’s active molecules, thus explaining the inability of DENV and ZIKV to initiate a productive infectious cycle in the host cells.

The results presented in this study underscore the entry antagonist as a promising class of antivirals and add *D. apetalum* to the group of medicinal plant extracts or phytochemicals that interfere with the early stage of the ZIKV replication cycle [11,12,20,41,42,43]. We recently reported that both extracts from *Aphloia theiformis* and *Psiloxylon mauritianum*, indigenous medicinal plants from Reunion Island, inactivate cell-free virions, as well as EGCG, impeding virus binding to the host cell surface by deforming ZIKV particles [12,20]. In addition, we recently described the mechanism of action of isoquercitrin (Q3G) that targets the internalisation process of ZIKV in different human cell lines [22]. In conclusion, we demonstrated that *D. apetalum* belongs to the growing list of Mascarene Islands plants which enable ZIKV and DENV inhibition at the virus entry level.

## 3. Materials and Methods

### 3.1. Cells, Viruses and Reagents

Human lung epithelial A549 cells (ATCC, CCL-185, Manassas, VA, USA), Vero cells (ATCC, CCL-81) and human-derived Huh-7 hepatoma cells (ATCC, PTA-8561) were grown in minimum essential medium (MEM: Gibco/Invitrogen, Carlsbad, CA, USA) supplemented with 10% heat-inactivated fetal bovine serum (FBS Good: Invitrogen), 2 mmol·L^−1^ L-Glutamine, 1 mmol·L^−1^ sodium pyruvate, 100 U·mL^−1^ of penicillin, 0.1 mg·mL^−1^ of streptomycin and 0.5 µg·mL^−1^ of fungizone (PAN Biotech) under a 5% CO_2_ atmosphere at 37 °C. The clinical isolate PF-25013-18 of ZIKV (ZIKV- PF13) and the recombinant Zika virus expressing the GFP reporter gene (ZIKV^GFP^) have been previously described [44,45]. Virus stocks were cultured and titrated on Vero cells using plaque-forming assay. DENV-1/FGA/89 was isolated in 1989 from a South American patient suffering from DF (GenBank: AF226687). DENV-2/ICC-265 was isolated from a DF patient in Brazil in 2009. DENV3/97 was isolated from a Dengue patient from north of Brazil in 2004 [46]. DENV-4/422 was isolated isolated in 2013 from a Brazilian Dengue patient with hemorrhagic manifestation [47]. The brazilian clinical isolate of Zika virus (ZK BR 2015/15261) was isolated from a patient with Zika fever from Northeast of Brazil in 2015. DENV stocks were grown in C6/36 cells and titrated by foci-forming immunodetection assay. Anti-pan flavivirus E monoclonal antibody 4G2-Alexa Fluor 594 was purchased from RD Biotech.

### 3.2. Extraction of Plant Material

Fresh aerial parts of *Doratoxylon apetalum* were collected in 2014, 2015 and 2017 in various locations of Reunion Island. Voucher specimens (RUN-055E) were deposited in the herbarium of the University of Reunion Island. Dried leaves (12.5 g) were reduced to powder and extracted by soaking in 50 mL ethanol:water (70:30) at room temperature for 24 h. Ethanol was then evaporated using a Rotavapor and aqueous phase was lyophilized using cryotec 20K (Cryotec, Saint-Gély-du-Fesc, France) to produce a brown powder. The crude extracts were solubilized in sterile phosphate buffer saline (PBS) and stored at –80 °C until used for the antiviral assays.

### 3.3. MTT Assay

Cells were cultured in a 96-well culture plate at a density of 1.5 × 10^4^ cells per well and treated with two-fold dilutions of plant extract ranging from 200 µg·mL^−1^ to 25 µg·mL^−1^. After a treatment period of 72 h at 37 °C, cells were rinsed with PBS 1× and 120 µL of culture medium mixed with 5 mg·mL^−1^ MTT (3-[4,5-dimethylthiazol-2-yl]-2,5- diphenyltetrahzolium bromide) solution was added to the cell monolayer. Incubation was extended for 2 h, then MTT medium was removed and the formazan crystals were solubilized with 50 µL of dimethyl sulfoxide (DMSO) [12]. Absorbance was measured at 570 nm with a background subtraction at 690 nm. The CC_50_ was determined using a nonlinear regression on Graphpad prism software.

### 3.4. Immunofluorescence and Flow Cytometry Assays

For cytometry assay, cells were harvested, fixed with 3.7% PFA in PBS for 20 min and subjected to a flow cytometric analysis using Cytoflex (Beckman Coulter, Brea, California, USA). Results were analyzed using cytexpert software. For immunofluorescence assay, cells grown on glass coverslips were fixed (PFA 3.7%) for 10 min, permeabilized for 4 min (PBS 1× 0.15% Triton X-100). Cells were stained for 1 h at room temperature in the dark for ZIKV using 4G2-Alexa 594 (1:1000 in PBS-BSA 2%). DAPI staining was used to delineate cell nuclei. Coverslips were mounted in Vectashield and the fluorescence was observed using Nikon Eclipse E2000-U microscope. Hamamatsu ORCA-ER camera and NIS-Element AR (Nikon, Melville, NY, USA) imaging software were used to capture images.

### 3.5. Virus Inactivation Assay

To assess the direct effect of the extract on viral infectivity, ZIKV^GFP^ (2 × 10^5^ plaque forming unit (PFU)) were mixed with extract at 200 µg·mL^−1^ and then incubated at 37 °C for 2 h. The mixture was diluted 50-fold (final virus concentration, 1 PFU/cell) in MEM containing 10% FBS to yield a subtherapeutic concentration of extract, and this mixture was subsequently added to A549 cells monolayer seeded in a 6-well plates. As a comparison, ZIKV^GFP^ was mixed with extract, diluted immediately to 50-fold (no incubation period) and added to cells for infection [12]. After 2 h of adsorption at 37 °C, the diluted inocula were discarded, and cells were washed with PBS twice. Medium overlay was applied and the plates were further incubated at 37 °C for 24 h. Cells and supernatants were collected and subjected to a flow cytometric analysis.

### 3.6. Western Blot

Cell lysates were performed in RIPA lysis buffer. All subsequent steps of immunoblotting were performed as previously described [22]. Primary antibodies were used at 1:1000 dilutions. Anti-mouse immunoglobulin-horseradish peroxidase conjugates was used as secondary antibody (dilution 1:500). Blots were revealed with ECL detection reagents.

### 3.7. Plaque Forming Assay

Viral progeny production was determined by plaque-forming assay as previously described with minor modifications [22]. Briefly, Vero cells grown in 48-well culture plate were infected by 0.1 mL of ten-fold dilutions of supernatants. Following an incubation of 2 h at 37 °C, 0.2 mL of culture medium supplemented with 5% fetal bovine serum (FBS) and 0.8% carboxymethylcellulose sodium salt (Sigma-Aldrich, Saint-Quentin-Fallavier, France) were added, and the incubation was extended for 4 days at 37 °C. The cells were fixed (PFA 3.7%) and stained with 0.5% crystal violet (Sigma-Aldrich) diluted in 20% ethanol, after the media had been removed. Plaques were counted and expressed as plaque-forming unit per mL (PFU·mL^−1^).

### 3.8. RT-qPCR

Total RNA including genomic viral RNA was extracted from cells with RNeasy kit (Qiagen) and reverse transcribed using E reverse primer (5’-TTCACCTTGTGTTGGGC-3’) and M-MLV reverse transcriptase (Life Technologies, Villebon-sur-Yvette, France) at 42 °C for 50 min. Quantitative PCR was performed on a CFX96 Real-Time PCR Detection System (Bio-Rad). Briefly, cDNA were amplified using 0.2 μM of each primer and GoTaq Master Mix (Promega, Charbonnières-les-bains, France). For each single-well amplification reaction, a threshold cycle (Ct) was calculated using the CFX96 program (Bio-Rad, Life Science, Marnes-la-Coquette, France) in the exponential phase of amplification. A synthetic gene coding for nucleotides 954 to 1306 of the MR766 strain (GenBank: LC002520) cloned in the pUC57 plasmid was used as template to generate a standard curve, which then served to make absolute quantitation of bound viruses.

### 3.9. Data analysis

Comparison between different concentrations was done by a one-way ANOVA test. All values were expressed as mean ± SD of at least three independent experiments. All statistical tests were done using the software Graph-Pad Prism (version 7.0; GraphPad software, La Jola, CA, USA). Degrees of significance are indicated on the figure as follows: * *p* < 0.05; ** *p* < 0.01; *** *p* < 0.001, **** *p* < 0.0001, *n.s.* = not significant.

## Figures and Tables

**Figure 1 ijms-20-02382-f001:**
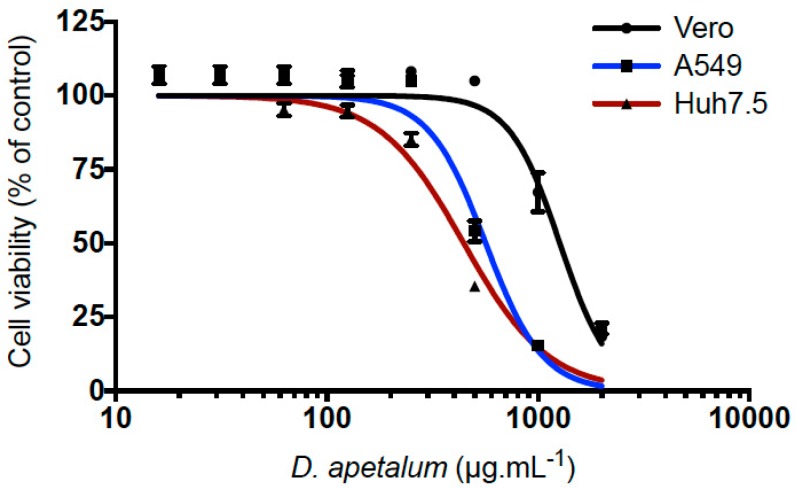
The cytotoxicity of *D. apetalum* extract on different cell lines. Vero, A549 and Huh7.5 cell lines were incubated with different concentrations of *D. apetalum* extract for 72 h. A 3-[4,5-dimethylthiazol-2-yl]-2,5- diphenyltetrazolium bromide (MTT) assay was performed to evaluate cell viability. Results are means ± SD of four independent experiments and are expressed as relative values compared to untreated cells.

**Figure 2 ijms-20-02382-f002:**
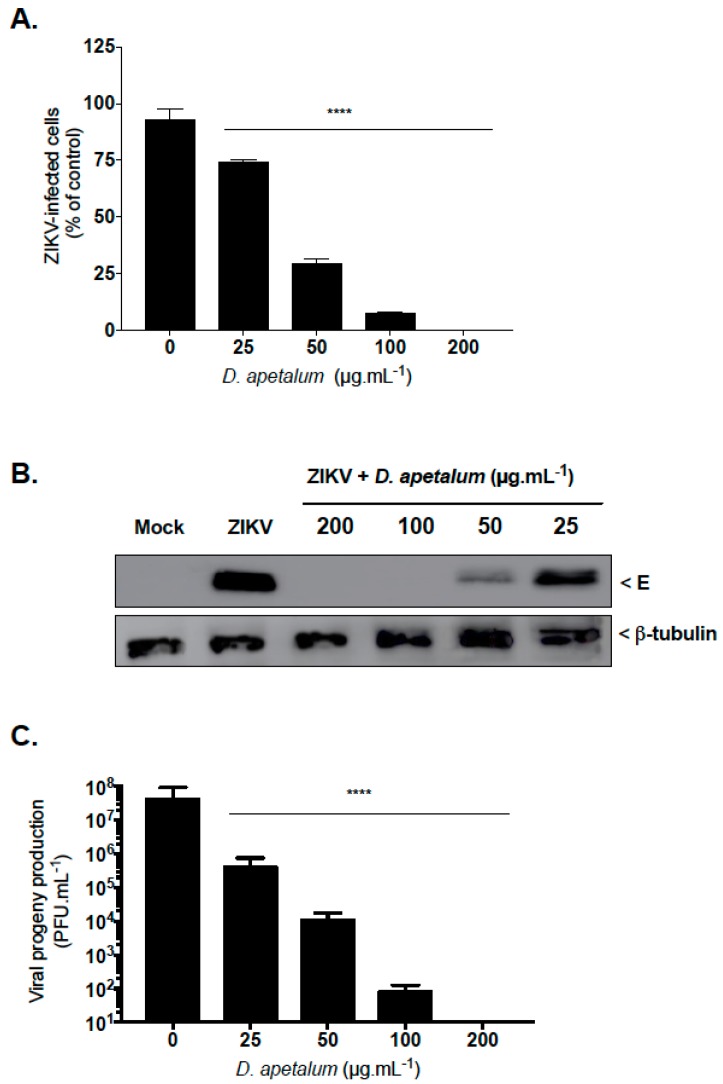
*D. apetalum* extract prevents infection of A549 cells by epidemic strain of Zika virus (ZIKV). A549 cells wer e infected with PF-25013-18 at a multiplicity of infection (MOI) of 2 and continuously incubated with different non-cytotoxic concentrations of *D. apetalum* extract. (**A**) Immunofluorescence analysis of viral protein expression in ZIKV-infected A549 cells. (**B**) Detection of intracellular E protein in ZIKV-infected A549 cells by immunoblot assay using anti-E mAb. β-tubulin served as loading control. (**C**) ZIKV progeny production was quantified by plaque-forming assay. Data represent the means ± SD from four independent experiments. One-way ANOVA and Dunnett’s test were used for statistical analysis (**** *p* < 0.0001).

**Figure 3 ijms-20-02382-f003:**
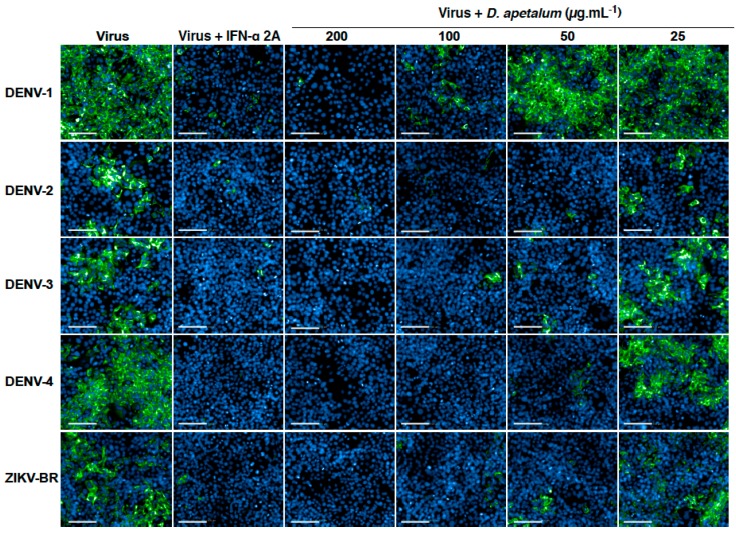
*D. apetalum* extract exhibits antiviral effect against the four Dengue virus (DENV) serotypes and an epidemic Brazilian strain of ZIKV. Huh7.5 cells were infected during 48 h with DENV-1 (MOI 2), DENV-2 (MOI 2), DENV-3 (MOI 0.5) or DENV-4 (MOI 2). Cells were infected for 48 h with the epidemic Brazilian strain (ZIKV-BR) of ZIKV at MOI 2. Infected Huh7.5 cells were continuously incubated with different non-cytotoxic concentrations of *D. apetalum* extract for 48 h. Recombinant IFN-α 2A (200 IU·mL^−1^) was added 2 h post infection and used as a positive control. Immunofluorescence assay was performed using anti-flavivirus E mAb 4G2. The percentage of immunostained cells was determined using the Operetta High-Content Imaging System. Scale bars are 100 µm.

**Figure 4 ijms-20-02382-f004:**
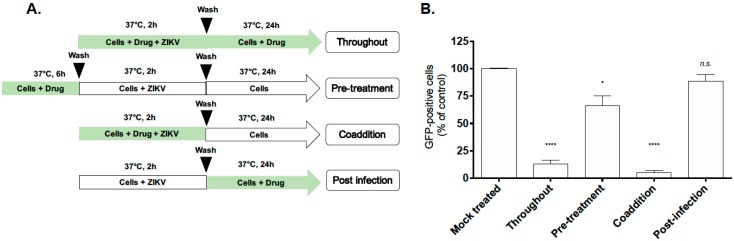
*D. apetalum* extract targets early stages of the ZIKV replication cycle. (**A**) Schematic representation of the time-of-drug-addition assay used to characterize antiviral activity of *D. apetalum* extract (200 µg·mL^−1^) on ZIKV^GFP^ infection of A549 cells. Green arrows indicate the presence of plant extract. (**B**) Results of GFP-expression in ZIKV^GFP^-infected A549 cells under different experimental conditions, shown in A, are analysed by flow cytometry assay. The data represent the means ± SD of four independent experiments and are expressed as relative values compared to the mock-treated control. One-way ANOVA and Dunnett’s test were used for statistical analysis (* *p* < 0.05; **** *p* < 0.0001; *n.s* = not significant).

**Figure 5 ijms-20-02382-f005:**
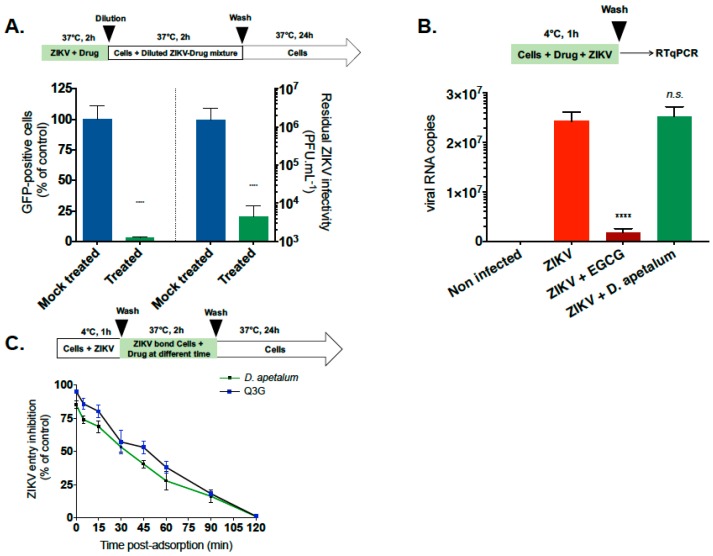
*D. apetalum* extract prevents ZIKV entry in A549 cells by inactivating virus particles. (**A**) Viral inactivation assay. ZIKV^GFP^ (2.10^5^ plaque forming unit (PFU)) was mixed with *D. apetalum* extract (200 µg·mL^−1^) for 2 h at 37 °C and then diluted 50-fold (final concentration, 1 PFU/cell) before infecting A549 cells. As a control, the same amount of virus was also mixed with *D. apetalum* extract but diluted immediately and applied to the A549 cells. Flow cytometric analysis of GFP fluorescence or viral titration using plaque forming assay were performed 24 h.pi. (**B**) A549 cells were infected with ZIKV at MOI of 1 for 1 h at 4 °C with or without 200 µg·mL^−1^ of *D. apetalum* extract. EGCG (100 µM) was used as positive control. The number of virus particles bound to the cell surface was measured by RT-qPCR. (**C**) A549 cells were incubated for 1 h with ZIKV^GFP^ at 4 °C. *D. apetalum* extract was added at different time points post temperature shift during 2 h. Q3G (200 µM) was used as a positive control. Data represent the means ± SD of three independent experiments performed in triplicate. One-way ANOVA and Dunnett’s test were used for statistical analysis (**** *p* < 0.0001; *n.s* = not significant).

**Table 1 ijms-20-02382-t001:** Cytotoxicity and antiviral activity of *D. apetalum* extract.

Virus	CC_50_ (µg·mL^−1^) ^a^	IC_50_ (µg·mL^−1^) ^b^	SI ^d^
DENV-1	263.5	96.35	2.7
DENV-2	299.0	16.75	17.8
DENV-3	293.0	25.90	11.3
DENV-4	303.0	23.30	13.0
ZIKV	295.5	17.50	16.8

Cytotoxic concentrations (CC_50_) and inhibitory concentrations (IC_50_) were obtained by performing nonlinear regression followed by the construction of sigmoidal concentration–response curves from Appendix A. ^a^ Concentration that inhibited cell viability by 50%; ^b^ concentration that inhibited infection by 50%; ^d^ selectivity index (CC_50_/IC_50_).

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
