# Peer review of "Doratoxylon apetalum, an Indigenous Medicinal Plant from Mascarene Islands, Is a Potent Inhibitor of Zika and Dengue Virus Infection in Human Cells"

_ijms, 2019, doi:10.3390/ijms20102382_

Round 1
Reviewer 1 Report
In this manuscript, Haddad et al describe the antiviral effects of the D. apetalum extract against ZIKV and DENV. The study design is quite similar to their previous publication testing a different plant extract against ZIKV and just as before, offers a potential mechanism of viral inhibition for the extract against ZIKV. This study is a promising start and warrants additional experiments that identify the specific antiviral components contained within the crude plant extract. Although it describes some interesting preliminary findings, the manuscript is not suitable for publication at this time. Overall, the manuscript requires major editorial assistance to correct errors in spelling, grammar and sentence structure (repeat use of passive voice). Additional concerns are listed below:
1. The title is phrased awkwardly – please rewrite. “Human” does not need to be capitalized.
2. Line 50 – is there a public health crisis currently ongoing for Zika virus? It appears that the number of cases has reduced dramatically since 2016. Please clarify.
3. Line 52-54: please cite.
4. In reference to lines 52-54: the authors have previously published a several studies highly relevant to this topic. Please provide additional background information about the types of compounds that have been tested, and describe the mechanism of action of these anti-ZIKV compounds.
5. Several sentences in the introduction section of the manuscript appear to be paraphrased from a previously published manuscript by the authors titled “Extract from Aphloia theiformis, an edible indigenous plant from Reunion Island, impairs Zika virus attachment to the host cell surface”. Given the high degree of experimental similarities between the two manuscripts, I would strongly encourage the authors to re-write the introduction section.
6. Lines 77-78: please change “no cytotoxic” to non-cytotoxic.
7. Lines 106-111: The IC50 and CC50 values were calculated based on a curve fit as described in the legend below Table 1. Some of the interpolated IC50 values fall outside the range of the concentrations tested by the authors. How can the authors be sure that IC50 values they report accurate if those concentrations were not even tested in their experiment? The only accurate way to describe the IC50 of DENV2, 3, 4 and ZIKV at this time would be as<25ug/mL.
If the authors want to emphasize that DENV2 is the most sensitive to the extract, they should repeat the experiment with additional concentrations below 25ug/mL and recreate the curve fit with the new data generated. I would strongly suggest the authors repeat the experiment to determine the correct IC50 values.
8. The text in the Results section describing Figure 5C is quite confusing. Please modify to indicate that 1) the extract was added at multiple time points after adsorption 2) the effect of each of these conditions was tested to determine the impact of the extract on viral internalization 3) the extract is most effective if added early during viral endocytosis. The current description includes phrases such as the extract “retained most of its antiviral activity”, which suggest that the extract may lose its antiviral activity if added at a later time point. This is not the case – it simply means that the extract is not effective if added post-internalization.
9. Please provide additional information about the ZIKV-GFP construct in the Methods section. Is any non-specific GFP signal observed in cells infected with this virus – meaning, does virus that has attached to the cell but not yet been internalized contribute to the overall GFP signal?
Author Response
Point-by-point responses to the referees
Reviewer 1:
Comments and Suggestions for Authors
In this manuscript, Haddad et al describe the antiviral effects of the D. apetalum extract against ZIKV and DENV. The study design is quite similar to their previous publication testing a different plant extract against ZIKV and just as before, offers a potential mechanism of viral inhibition for the extract against ZIKV. This study is a promising start and warrants additional experiments that identify the specific antiviral components contained within the crude plant extract. Although it describes some interesting preliminary findings, the manuscript is not suitable for publication at this time. Overall, the manuscript requires major editorial assistance to correct errors in spelling, grammar and sentence structure (repeat use of passive voice). Additional concerns are listed below:
Please accept our apologies for the grammatical errors and sentences structure.
The grammar errors and sentences within the main text, graphs and figure legends have been modified and corrected accordingly.
1. The title is phrased awkwardly – please rewrite. “Human” does not need to be capitalized.
We do apologize for this error. Title has rewritten accordingly. Lines 2-4.
2. Line 50 – is there a public health crisis currently ongoing for Zika virus? It appears that the number of cases has reduced dramatically since 2016. Please clarify.
We do apologize for this imprecise information. Thus, sentence has been modified accordingly. Lines 49-51.
3. Line 52-54: please cite.
Citations have been added, Line 54.
4. In reference to lines 52-54: the authors have previously published a several studies highly relevant to this topic. Please provide additional background information about the types of compounds that have been tested, and describe the mechanism of action of these anti-ZIKV compounds.
We agree with the reviewer 1. We briefly described the mechanism of action of anti-ZIKV compounds (Q3G and EGCG) which have been identified as antiviral compounds by our team (Gaudry et al., 2018 ; Clain et al., 2018) and others (Carneiro et al., 2016 ; Sharma et al., 2017).
The main text has been modified accordingly (lines 56-60)
5. Several sentences in the introduction section of the manuscript appear to be paraphrased from a previously published manuscript by the authors titled “Extract from Aphloia theiformis, an edible indigenous plant from Reunion Island, impairs Zika virus attachment to the host cell surface”. Given the high degree of experimental similarities between the two manuscripts, I would strongly encourage the authors to re-write the introduction section.
We do apologize that the introduction section is paraphrased from our previously manuscript, Thus, we modified the introduction accordingly.
6. Lines 77-78: please change “no cytotoxic” to non-cytotoxic.
We do apologize for this error. Change has been made accordingly. Line 81
7. Lines 106-111: The IC50 and CC50 values were calculated based on a curve fit as described in the legend below Table 1. Some of the interpolated IC50 values fall outside the range of the concentrations tested by the authors. How can the authors be sure that IC50 values they report accurate if those concentrations were not even tested in their experiment? The only accurate way to describe the IC50 of DENV2, 3, 4 and ZIKV at this time would be as<25ug/mL.
If the authors want to emphasize that DENV2 is the most sensitive to the extract, they should repeat the experiment with additional concentrations below 25ug/mL and recreate the curve fit with the new data generated. I would strongly suggest the authors repeat the experiment to determine the correct IC50 values.
We thank reviewer 1 for raising this point and we agree with the reviewer 1 on this matter. Thus, in the revised version of the manuscript, a new range of concentration of plant extract (with additional concentrations below 25 µg.mL-1) have been added in order to evaluate the correct IC50.
Thus, in the revised version of the manuscript, figures S1 and Table 1 have been modified accordingly.
8. The text in the Results section describing Figure 5C is quite confusing. Please modify to indicate that 1) the extract was added at multiple time points after adsorption 2) the effect of each of these conditions was tested to determine the impact of the extract on viral internalization 3) the extract is most effective if added early during viral endocytosis. The current description includes phrases such as the extract “retained most of its antiviral activity”, which suggest that the extract may lose its antiviral activity if added at a later time point. This is not the case – it simply means that the extract is not effective if added post-internalization.
We do apologize for this unclear section concerning the internalisation process inhibited by D. apetalum extract.
As suggested by the reviewer 1, we modified the sentences accordingly. Lines 180-193.
9. Please provide additional information about the ZIKV-GFP construct in the Methods section. Is any non-specific GFP signal observed in cells infected with this virus – meaning, does virus that has attached to the cell but not yet been internalized contribute to the overall GFP signal?
The design of recombinant Zika virus expressing reporter protein eGFP has been described elsewhere (Gadea et al., 2016, ref 45). Expression of eGFP is only detected during the steps of ZIKV RNA replication in infected cells by ZIKV-GFP. Virus particles are composed of structural proteins C, M, and E surrounding the genomic viral RNA. No eGFP molecules could be detected within the virus particles thus excluding the possibility that ZIKV-GFP shows GFP-mediated auto-fluorescence.
Reviewer 2 Report
This is an interesting study that uncovered the antiviral activity of Doratoxylum apetalum against ZIKV and Dengue by preventing the internalization of virus particles into the host cells. But, additional experiments need to be performed to support their conclusions.
Major concerns in the manuscript:
As a mechanism, the authors suggested that Doratoxylum apetalum is preventing the internalization of virus particles (Fig5A). But, this conclusion need to be supported by additional experiments. There are several ways to further investigate this mechanism and confocal microscopy would help for it. The following suggestions are just to give them an example.
a) Since viral attachment is no affected (Fig. 5B), authors could track the particle during the entry and show that the internalization is prevented by confocal microscopy.
b) In the case there is internalization, authors could check whether the endosomal pH is affected and chloroquine could be used as a positive control.
c) Author could also investigate the antiviral effect on particle release from the endosome into the cytoplasm.
This reference (PMID: 30404919) could potentially help the authors to find details about some protocols.
Minor concerns in the manuscript:
- The authors showed a strong evidence that the viral attachment is not affected by binding assay (Fig 5B). This Figure 5B can be moved to be 5A.
- In the line 269, need to start with capital letter.
Author Response
Reviewer 2
Comments and Suggestions for Authors
This is an interesting study that uncovered the antiviral activity of Doratoxylum apetalum against ZIKV and Dengue by preventing the internalization of virus particles into the host cells. But, additional experiments need to be performed to support their conclusions.
Major concerns in the manuscript:
As a mechanism, the authors suggested that Doratoxylum apetalum is preventing the internalization of virus particles (Fig5A). But, this conclusion need to be supported by additional experiments. There are several ways to further investigate this mechanism and confocal microscopy would help for it. The following suggestions are just to give them an example.
a) Since viral attachment is no affected (Fig. 5B), authors could track the particle during the entry and show that the internalization is prevented by confocal microscopy.
b) In the case there is internalization, authors could check whether the endosomal pH is affected and chloroquine could be used as a positive control.
c) Author could also investigate the antiviral effect on particle release from the endosome into the cytoplasm.
This reference (PMID: 30404919) could potentially help the authors to find details about some protocols.
We appreciated the remarks raised by the reviewer 2. Additional experiment was performed showing that D. apetalum and isoquercitrin (Q3G) have a similar time-course of ZIKV inhibition. Indeed, we previously demonstrated that Q3G exerts antiviral effect against ZIKV at the virus internalisation level (IJMS, PMID:29621184). Time-course analysis of Q3G-mediated ZIKV inhibition showed that one-half hour after the initiation of virus entry process (post-binding step in our protocol) into the host-cell, ZIKV-GFP was largely insensitive to Q3G antiviral action.
A new figure 5C was added in the revised manuscript and the text was modified accordingly. Concerning the confocal microscopy analysis, in visu experiments will be undertaken for the monitoring of ZIKV entry in host-cells in presence of D. apetalum. We apologize for not being able to provide such information in this current revised manuscript. However, we would be pleased to submit our future data in a next manuscript.
Further experiments using confocal microscopy could be undertaken to validate our assumption that D. apetalum-mediated ZIKV inhibition essentially relates to a lack of virus internalization into the host-cell (reference 39 has been added).
Minor concerns in the manuscript:
- The authors showed a strong evidence that the viral attachment is not affected by binding assay (Fig 5B). This Figure 5B can be moved to be 5A.
Since the time-of-drug-addition assay showed that the early stage of ZIKV infection is inhibited, we first wondered to evaluate If D. apetalum extract acts directly on ZIKV (cell-free virions) infectivity before studying the effect on the binding step.
- In the line 269, need to start with capital letter.
We do apologize for this mistake, change has been made.
Round 2
Reviewer 1 Report
While the authors have incorporated several changes suggested by the reviewers, the revised manuscript is still littered with grammatical and spelling mistakes. Few examples include:
1) "falvivirus" instead of flavivirus in line 82
2) Line 49: "At present, there is no therapeutics licensed against..." Should be there "are".
3) Repeat use of the word "recent" or "recently" in lines 56-60
4) Line 60: "Recent studies have been demonstrated" Here there is no need for the word been in the sentence
5) there should be an "of" after the word regardless in line 69
These are just a few examples, three of which are in the introduction section alone. The authors should read over the manuscript in its entirety carefully and make the necessary revisions to improve the quality of writing.
Author Response
Reviewer 1:
Comments and Suggestions for Authors
1) "falvivirus" instead of flavivirus in line 82
We do apologize for this mistake, change has been done.
2) Line 49: "At present, there is no therapeutics licensed against..." Should be there "are".
We do apologize for this mistake, change has been done.
3) Repeat use of the word "recent" or "recently" in lines 56-60
Modifications have been done.
4) Line 60: "Recent studies have been demonstrated" Here there is no need for the word been in the sentence
We do apologize for this error, change has been done.
5) there should be an "of" after the word regardless in line 69
We do apologize for this error, addition of “of ” has been done.
Please accept our apologies for the grammatical errors and spelling mistakes. The grammar errors and within the main text have been carefully modified and corrected accordingly.
Reviewer 2 Report
I am happy with the new figure 5C provided in the revised manuscript.
Author Response
Reviewer 2:
I am happy with the new figure 5C provided in the revised manuscript.
We thank the reviewer for his/her positive appreciation of our work.